

# Whispers from the dark side:
# Confronting light new physics with NANOGrav data

**Wolfram Ratzinger⋆ and Pedro Schwaller†**

PRISMA+ Cluster of Excellence & Mainz Institute for Theoretical Physics,
Johannes Gutenberg-Universität Mainz, 55099 Mainz, Germany

⋆ w.ratzinger@uni-mainz.de, † pedro.schwaller@uni-mainz.de

## Abstract

The NANOGrav collaboration has recently observed first evidence of a gravitational wave background (GWB) in pulsar timing data. Here we explore the possibility that this GWB is due to new physics, and show that the signal can be well fit also with peaked spectra like the ones expected from phase transitions (PTs) or from the dynamics of axion like particles (ALPs) in the early universe. We find that a good fit to the data is obtained for a very strong PT at temperatures around 1 MeV to 10 MeV. For the ALP explanation the best fit is obtained for a decay constant of $F \approx 5 \times 10^{17}$ GeV and an axion mass of $2 \times 10^{-13}$ eV. We also illustrate the ability of PTAs to constrain the parameter space of these models, and obtain limits which are already comparable to other cosmological bounds.

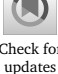
## 1 Introduction

With the first direct observation of gravitational waves (GWs) by LIGO [1], a new era in astrophysics and cosmology has started. Since GWs travel almost undisturbed through spacetime,

they can carry information from before the time of CMB emission, which is where our direct observations using electromagnetic radiation end. GWs therefore open a new window to the early Universe.

Pulsar Timing Arrays such as EPTA [2], PPTA [3] and NANOGrav [4] are sensitive to GWs with frequencies of $10^{-8}$ Hz and below. A stochastic background of such low frequency GWs could be produced in the early universe by a variety of processes, such as inflation, cosmic strings, phase transitions, or scalar field dynamics [5]. The most recent data release of the NANOGrav collaboration [6] for the first time shows evidence for such a stochastic GW background, which is well described by a $f^{-2/3}$ power law spectrum with a GW strain amplitude of $2 \times 10^{-15}$, or equivalently a GW energy density $\Omega_{\mathrm{GW}} h^2$ of order $10^{-10}$.[1] This is indeed consistent with the GW density one expects from a variety of cosmological sources, as was discussed for the case of cosmic strings [7–9], phase transitions [10, 11], or primordial black hole formation [12, 13].

So far these studies have focussed on demonstrating that a sufficiently large GW density can be achieved in these models in the required frequency range. Here we perform the first fit to the frequency binned NANOGrav data. Since most cosmological sources of GWs have specific spectral features, it is important to verify that indeed they agree well with the data. In doing this, we are able to obtain best fit parameter regions for two classes of models that produce primordial GWs, namely phase transitions in the early universe [14–18] and audible axions [19–21]. We also show that the NANOGrav data already puts constraints on the parameter space of these models, which are comparable to the ones coming from other astrophysical observations such as big bang nucleosynthesis (BBN) or the constraint on the number of relativistic degrees of freedom, $N_{\mathrm{eff}}$.

With more precise data it will become possible to distinguish between different cosmological sources and from the expected background due to supermassive black hole binaries. Our work presents a first step in this direction. It is organised as follows: In the next section, we describe our effort at recasting the NANOGrav data, and re-derive the best fit regions for single power law fits. The following two sections introduce the parameterisation of the stochastic GW background produced by audible axions and phase transitions, respectively, and the best fit regions for the model parameters, before we present our conclusions.

## 2 Refitting the NANOGrav data

The magnitude of a stochastic GW background is typically described by the dimensionless, frequency dependent characteristic strain amplitude $h_c(f)$. For a single power law it can be written as

$$h_c(f) = A_{\mathrm{GW}} \left( \frac{f}{f_y} \right)^{\alpha}, \tag{1}$$

where $A_{\mathrm{GW}}$ is the amplitude, $\alpha$ is the slope and $f_y = 1/\mathrm{year}$ is a reference frequency at which the amplitude is fixed. An important related quantity is the energy density in GWs as a fraction of the critical energy density, $\Omega_{\mathrm{GW}}$, which is given by [4]

$$\Omega_{\mathrm{GW}}(f) h^2 = \frac{2\pi^2}{3 H_{100}^2} f^2 h_c^2(f), \tag{2}$$

where $H_{100} = 100$ km/s/Mpc and $H_0 = h H_{100}$ is the Hubble rate today with $h \approx 0.7$.

---

[1] The NANOGrav collaboration so far hesitates to claim the detection of a stochastic GW background, since they have not observed the characteristic quadrupolar signature with sufficient significance yet.

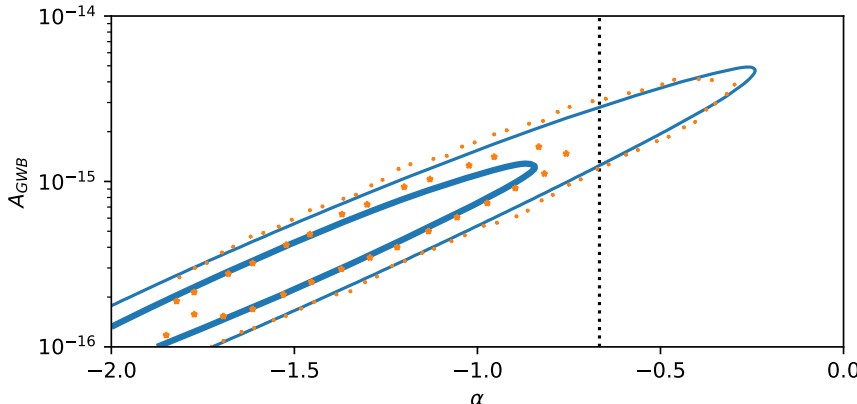

Figure 1: Comparison of $1\sigma$ and $2\sigma$ contours for a single power law fit to the 5 lowest frequency bins. Our results are shown with continuous blue lines and the original result with orange dots. The black dotted line sits at $\alpha = -2/3$, the expected slope for the signal of SMBHs.

In Fig. 1 of Ref. [6] the NANOGrav collaboration provides the results of different fits to the data, namely a free spectrum fit of the individual frequency bins, a fit of a single power law to the lowest 5 frequency bins or to all 30 bins, and a broken power law with different slopes for the low and high frequency part of the data. The high frequency bins are expected to be dominated by white noise with slope $\alpha = 3/2$, which is corroborated by the broken power law fit. Instead the 5 lowest frequency bins contribute 99.98% of the significance of the potential GW signal.

In the following, we will therefore fit our signal models to the 5 lowest frequency bins, assuming that the remaining data points are explained by white noise. The results of the free spectrum fit are given in terms of the timing residual, which is related to the characteristic strain as

$$\text{residual}(f) = \frac{1}{4\pi^2 f_y}\left(\frac{f}{f_y}\right)^{-3/2} h_c(f),\tag{3}$$

in units of seconds. Note that we have chosen the prefactor in this formula such that by fitting a single power law to the data, we can reproduce the best fit contours of [6], see Fig. 1. In the following sections, we will fit this data with signal templates motivated by concrete new physics scenarios.

## 3 Audible axions and NANOGrav

The audible axion is a simplified model where an axion-like particle $a$ couples to a dark photon $X$ through a term of the form

$$\mathcal{L} \supset -\frac{q}{4F} a X_{\mu\nu}\tilde{X}^{\mu\nu},\tag{4}$$

where $F$ is the axion decay constant, i.e. the scale where the global symmetry in the UV is broken and gives rise to the light pseudoscalar $a$, $q$ is a dimensionless charge, and $X_{\mu\nu}$ and $\tilde{X}_{\mu\nu}$ are the dark photon field strength tensor and its dual. The axion has a potential $V(a) = m_a^2 F^2 (1 - \cos(a/F))$, such that its mass is given by $m_a$.

As usual in the axion misalignment mechanism, we assume that after the end of inflation, the axion is displaced from the minimum of $V(a)$ by $\theta F$, with $\theta$ an order one angle. The axion remains displaced until the Hubble rate becomes of order $m_a$, at which point it starts to oscillate around the origin. It was shown in [22–24] that the presence of a dark photon leads to a suppression of the axion dark matter abundance, making larger values of $F$ consistent with observations. An efficient energy transfer to the dark photons is possible due to a tachyonic instability that develops while the axion rolls. The same process also amplifies quantum fluctuations in the dark photon field, which grow to macroscopic scales and source a detectable GW background [19].

The GW spectrum produced by audible axions is peaked at the frequency corresponding to the dark photon momentum mode that grows the fastest, and is closely related to the axions mass $m_a$. In terms of the model parameters, the peak frequency, redshifted to today, can be estimated as

$$f_0^{\text{peak}} \approx 1.1 \times 10^{-8} \text{ Hz} \left(\frac{q\theta}{50}\right)^{\frac{2}{3}} \left(\frac{m_a}{10^{-12}\,\text{meV}}\right)^{\frac{1}{2}}. \tag{5}$$

The amplitude of the GW signal is determined by the strength of the source, i.e. the energy that is initially carried by the axion. This is mostly influenced by the size of the decay constant $F$. The peak amplitude of the signal can be estimated as

$$\Omega_{\text{GW}}^0 h^2 \approx 1.84 \times 10^{-7} \left(\frac{F}{m_{pl}}\right)^4 \left(\frac{\theta^2}{q} 50\right)^{4/3}. \tag{6}$$

To perform our fits we use the signal shape provided in [20]

$$\Omega_{\text{GW}}^0(f)h^2 = \Omega_{\text{GW}}^0 h^2 \frac{6.3 \left(f/(2f_0^{\text{peak}})\right)^{3/2}}{1 + \left(f/(2f_0^{\text{peak}})\right)^{3/2} \exp\left[12.9 \left(f/(2f_0^{\text{peak}}) - 1\right)\right]}. \tag{7}$$

In Fig. 2 we show on the left the best fit of an audible axion compared to the five first frequency bins from NANOGrav. On the right we show the one and two sigma contours in the $F$-$m_a$ plane with $\theta = 1$ and $q = 50$ fixed. To get such a strong signal the energy in the axion that is transmitted to the dark photon has to be quite significant. The dark photon is a form of dark radiation and therefore contributes to the number of relativistic degrees of freedom $N_{\text{eff}}$. From Fig. 2 it becomes clear that this excludes approximately half of the parameter space in the best fit region. Since the emission of GWs only proceeds at a temperature of a few MeV, close to the onset of BBN, it seems challenging to further reduce the dark photon abundance to avoid this bound.

Values of $F$ and $m_a$ which lie above the green contours predict a GW signal which is too large, i.e. this region is excluded by the NANOGrav data. While the $N_{\text{eff}}$ is slightly stronger, it is worth noting that PTAs are already able to put competitive bounds on this scenario.

In the left panel of Fig. 2 we show the best fit from the Audible Axions model together with the best fits from phase transitions discussed hereafter. As one can see, the sharp exponential decrease in signal strength predicted by the Audible Axions model is hard to accommodate by the data. Our analysis suggests that the Audible Axion model is disfavored by a Bayes factor of $\approx 800$ compared to the simple power law, while it only mildly disfavors the non-runaway PT by a factor of $\approx 2$ and even favors the runaway PT by a factor of $\approx 4$ compared to the power law. These results have to be taken with a pinch of salt however, since our analysis does not consistently include all the different noise sources. We leave a proper study of the potential to discriminate models with the full NANOGrav data for future work.

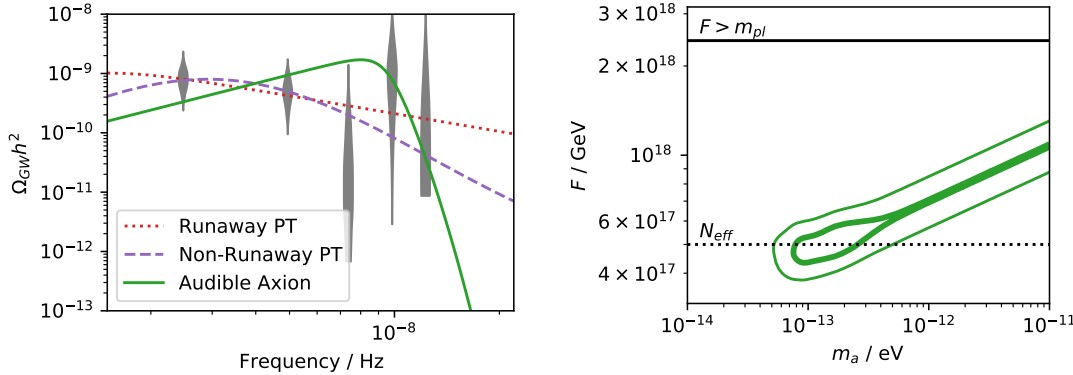

Figure 2: Left: Signal of the best fits of a runaway and a non-runaway phase transition as well as an audible axion compared to the first frequency bins of NANOGrav in the frequency-$\Omega_{GW}h^2$ plane. Right: $1\sigma$ and $2\sigma$ regions in the $F$-$m_a$ plane parameterizing the audible axion. The horizontal lines indicate the bounds originating from the decay constant $F$ having to be smaller than the Planck mass $m_{pl}$ and from the dark photon relic density not violating the bounds on $N_{\text{eff}}$.

## 4 Phase transitions and NANOGrav

It has been known for many years that a cosmological phase transition (PT), such as from the spontaneous breaking of a global or gauge symmetry through a scalar field that acquires a vacuum expectation value, produces a stochastic GW background if the transition is strongly first order [14–16]. While a large variety of models exists that predict such a transition at different scales, the GW signal of a strong first order PT is universally described by only four parameters, the ratio between the vacuum and total energy density $\alpha = \rho_{\text{vac}}/\rho_{\text{tot}}$, the time scale of the transition $\beta/H$, where $H$ is the Hubble scale at the time of the transition, the temperature $T_*$ at which the transition takes place and the bubble wall velocity $v_w$ [17,25].[2]

We use the signal templates in terms of these parameters as given in [26]. The peak frequencies and amplitudes of the two most important contributions to the signal scale as

$$f_p \approx 2 \times 10^{-7}\text{Hz}\left(\frac{\beta}{H}\right)\left(\frac{T_*}{\text{GeV}}\right), \tag{8}$$

$$\Omega_{\text{GW}}h^2 \approx 10^{-6}v_w\left(\frac{\beta}{H}\right)^{-n}\left(\frac{\alpha}{1+\alpha}\right)^2, \tag{9}$$

where $n = 1$ for the sound wave contribution and $n = 2$ for the scalar field contribution, and we neglect order one numbers which are not relevant for the qualitative discussion. Very strong transitions are characterised by $\alpha > 0.1$ and a wall speed approaching the speed of light, $v_w \to 1$. The NANOGrav signal corresponds to an energy density $\Omega_{\text{GW}}h^2 > 10^{-10}$ at a frequency around $10^{-8}$ Hz, so that only a strong transition will be able to explain the data. Furthermore we immediately see that $T_*$ should be of order $10^{-3}-10^{-2}$ GeV, i.e. the PT should happen at a very low scale. The implications of this for concrete models will be discussed in more detail below.

We consider two scenarios. If the PT takes place at a temperature significantly below the critical temperature, the Universe will be dominated by vacuum energy, i.e. the $\alpha$ dependence

---

[2]The PT might potentially proceed in a dark sector, in which case the SM and dark sector temperature can differ. In this work we have for simplicity always assumed that the dark sector temperature at the time of the PT is equal to the SM temperature.

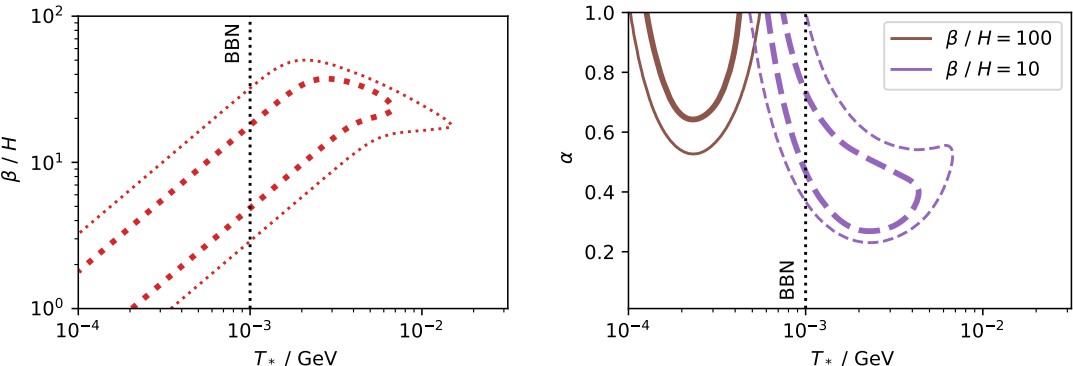

Figure 3: Left: Regions favoured by the NANOGrav signal for a vacuum PT, with $v_w = 1$, shown as a function of the transition temperature $T_*$ and the PT timescale $\beta/H$. Right: Same for a strong first order PT in a plasma, with $v_w = 1$ and fixed values of $\beta/H$, as function of $T_*$ and the energy budget $\alpha$. The vertical line at one MeV indicates the onset of BBN, below which strong constraints apply to any models that alter the expansion rate of the Universe.

drops out of Eq. (9). In such a supercooled PT, no friction acts on the bubble wall, so that $v_w = 1$. Furthermore in the absence of a plasma, the only source of GWs is the scalar field itself, i.e. $n = 2$ in Eq. (9), and the GW signal shape is best described by the envelope approximation [17]. In that case, a good fit to the data requires relatively small values of $\beta/H \lesssim 50$, and transition temperatures around or below the MeV scale, as shown in Fig. 3. Above the peak frequency, the GW strain amplitude of the PT signal falls as $f^{-3/2}$. Therefore if the peak frequency lies below the lowest frequency probed by NANOGrav, the signal will look like a single power law to the detector. This explains the flat direction in the fit towards lower temperatures and lower values of $\beta/H$. However lower values of $\beta/H$ are increasingly difficult to obtain in realistic models, therefore this region should be considered less favoured.

If the PT is very strong but not supercooled, the bubble walls will still reach a relativistic terminal velocity, so for simplicity we again set $v_w = 1$. In this case sound waves in the plasma induced by the PT are the dominant source of GWs, and the amplitude is only suppressed by one power of $\beta/H$. As expected, in Fig. 3 we see that a good fit to the data in the $T_*-\alpha$ plane is found both for $\beta/H = 10$ and $\beta/H = 100$, where in the second case the suppression of the signal is compensated by a larger energy budget $\alpha$. Again we also find a flat direction, where the peak of the PT signal is shifted below the NANOGrav frequency range, and data is fit by the high frequency tail. The flat direction is however smaller than for the vacuum dominated case, since the steep decrease of the strain sourced by sound waves beyond the peak $\propto f^{-3}$ is disfavored by the data.

In both scenarios, we find that the PT should happen at a temperature around 1 MeV, with only a small viable region slightly above 10 MeV. Since extensions of the SM at such low scales are almost impossible to hide from laboratory experiments, it is clear that the PT should take place in a dark sector, with only very weak interactions with the SM [26–33].

Nevertheless it was shown in [26] that also PTs in a dark sector are subject to strong constraints, in particular if they happen close to the scale of BBN. The reason is that BBN is a sensitive probe of the Hubble scale at temperatures below the MeV scale, which in turn depends on the total energy density in the Universe, since gravity is universal. Either the energy density in the hidden sector should be transferred to the SM before the onset of BBN at $T \sim 1$ MeV, which essentially prohibits PTs below that scale, or the energy should be converted into dark radiation, in which case the dark sector temperature is constrained by $N_{\text{eff}}$.

Viable models should therefore have few degrees of freedom, and still feature a very strong first order PT. The simplest scenario is probably a single scalar field with a non-renormalizable potential, such as a very light radion or dilaton. Indeed for these models it is known that a strongly supercooled first order PT can occur and produce a large GW background [34–38]. For renormalizable scenarios, the most minimal models that were found in [26] consist of either two real singlet scalars or a $U(1)$ gauge boson with a complex scalar charged under the gauge symmetry. While the majority of the parameter space of these models features a weaker PT, there are benchmark points with $\alpha > 0.5$ and $\beta/H \lesssim 100$, while still being consistent with constraints from BBN or $N_{\text{eff}}$.

Finally also here it should be noted that PTs with $T_* \sim 1$ MeV which produce a GW signal stronger than the observed one are now excluded by the NANOGrav data. We are therefore finding the first non-trivial constraints on the dynamics of potential dark sectors around these scales. Of course, to obtain robust limits on concrete models, a reduction of the large theoretical uncertainties in the prediction of the GW signals would be desirable. In particular new results for the sound wave spectrum keep appearing [39–43]. While these could slightly shift the contours in Fig. 3, the overall picture remains unchanged. For other recent progress, see e.g [40, 44–46].

# 5 Discussion and Outlook

The first hint of a GWB observed by NANOGrav is very intriguing. While the data can be well explained with a single power law, consistent with the expected background from supermassive black hole binaries (SMBHBs), we show here that also broken power law spectra, which are predicted in various extensions of the SM, can well describe the signal.

In both new physics scenarios we considered, the peak of the GW signal is strongly correlated with the relevant mass scale of the new physics, either the axion mass or the mass scale of the new sector that undergoes a phase transition. The PTA data therefore already allows us to narrowly constrain the potential mass range.

Since the data suggests very light new physics, it is already clear that these new particles have to be part of a dark sector that is only very weakly coupled to the SM, otherwise laboratory experiments would have uncovered them already. Yet astrophysical data on BBN and $N_{\text{eff}}$ constrain the parameter space of such dark sectors.

For the audible axion scenario, we find parameter regions consistent with $N_{\text{eff}}$ for masses around $10^{-13}$ eV and a decay constant of $5 \times 10^{17}$ GeV. This region may be probed in the future by the CASPEr-wind experiment [47], and also by future black hole binary merger data through the superradiance effect [48].

A first order PT can explain the data if the transition is very strong and happens at temperatures between 1-10 MeV, or slightly below, if BBN and $N_{\text{eff}}$ constraints can be evaded. We have briefly illustrated some dark sector models that are known to satisfy all requirements. Here it will of course be interesting to ask whether concrete realisations can also explain the observed dark matter abundance, and whether they leave observable imprints elsewhere.

Already this first hint of a stochastic GW background in the PTA range provides us with a deep insight into possible new physics explanations of the signal. With more precise frequency binned data it will be possible to distinguish between different models and astrophysical backgrounds such as the one from SMBHBs. It would also be interesting to directly fit a broader range of GW templates to the pulsar timing data, possibly including polarised signals such as the one expected from audible axions. Exciting times lie ahead!

# Acknowledgments

We would like to thank Moritz Breitbach, Michael Geller, Joachim Kopp, Eric Madge, Lisa Michaels, Toby Opferkuch, Daniel Schmitt and Ben Stefanek for useful discussions. Our work is supported by the Deutsche Forschungsgemeinschaft (DFG), Project ID 438947057. We also acknowledge support by the Cluster of Excellence "Precision Physics, Fundamental Interactions, and Structure of Matter" (PRISMA+ EXC 2118/1) funded by the German Research Foundation (DFG) within the German Excellence Strategy (Project ID 39083149).

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
