# Peer review of "Whispers from the dark side: Confronting light new physics with NANOGrav data"

_SciPost Physics, doi:SciPost Phys. 10, 047 (2021)_

## Round 1 · Referee Report · Anonymous (Referee 1) · 2020-11-10

Strengths

See report.

Weaknesses

See report.

Report

The authors discuss the interpretation of the recent NANOGrav results in the context of ALPs and cosmological phase transitions. The main finding is that in principle both scenarios could be responsible for the signal.

The analysis is easy to follow and seems sound. I only have a couple of minor issues that the authors should improve upon.

It is by no means clear that the signal seen by NANOGrav are gravitational waves. In fact the paper [4] discloses that the signal has not the typical form from a quadrupole that is expected for gravitational waves. Hence, it is in fact rather unlikely that the signal comes from gravitational waves. The author should point this out in the introduction.

Even though the authors provide the best-fit points and the 1- and 2-sigma contours, they do not provide the goodness of the fit in the best-fit point. Which of the different scenarios they describe provides the best fit to the data (in relation to the number of free parameters in the model, and so on)?

One central ingredient in the part on the phase transition is the spectral shape. The authors use that for large frequencies, the spectrum scales as f^-3/2. The authors should disclose where this scaling is coming from ([25] seems to quote a different scaling ~f^-4 for Omega).

In summary, the authors should improve on above points. After that I would recommend publication in SciPost Physics.

Requested changes

See report.

  • validity: high
  • significance: high
  • originality: good
  • clarity: high
  • formatting: excellent
  • grammar: excellent

Author:  Pedro Schwaller  on 2020-11-13  [id 1045]

(in reply to Report 1 on 2020-11-10)

We thank the referee for the positive evaluation of our manuscript. Here are our responses to the comments:

It is true that NANOGrav has not claimed the observation of GWs since the quadrupole nature of the signal could not yet be established with high significance, and we will add a footnote to the introduction to clarify this.
We do however not share the pessimistic view that it is unlikely that these are GWs, after all a GW signal in that region is expected from astrophysical sources (supermassive black hole binaries). As we discuss in section V, an interesting task for the future will be whether a potential primordial signal such as the one proposed in our work can be distinguished from these backgrounds.

We will add a paragraph at the end of section III discussing the Bayes factors of the different models. Compared with fitting a single power law, one phase transition scenario provides a better fit for the data, while the second phase transition scenario and the axion model perform somewhat worse. However we want to stress that a full fit to the NANOGrav timing data should be done to accurately assess how good a particular model fits the data.

The GW signal for vacuum PTs is described by the envelope approximation, for which Omega scales as f^-1 for large frequencies, and thus the strain behaves as f^-3/2. This is explained e.g. in Ref. [17], and was not re-discussed in Ref. [25] since no significant new insights into this scenario happened in the last 4 years. We will add a sentence and Ref [17] on page 6 to make this more clear.
For the second PT scenario, the signal falls more steeply at large frequencies (Omega ~ f^-4, strain ~ f^-3). This explains why the runaway PT scenario gives a slightly better overall fit to the data compared with the non-runaway scenario.

---

## Round 1 · Referee Report · Anonymous (Referee 2) · 2020-11-20

Strengths

1- Interesting analysis of potential physics implications of new experimental result with two different models 2- Main ideas and results are clearly presented

Weaknesses

1- Lack in details when comparing preferred parameter space to existing constraints for both models 2- No discussion of how the two models compare to each other in terms of fitting the data

Report

The authors show that the NANOGrav result can be explained by gravitational waves generated in two beyond the standard model scenarios: with axion-like particles (ALPs); and with new first order phase transitions (PT). They also perform the first fit to NANOGrav data with BSM models using the binned frequency data instead of just the overall size of the signal. Their manuscript is interesting and presents new results.

One of their main conclusions is that in both cases the signal must be generated at temperatures close to MeV, which poses significant challenges due to constraints from Big Bang Nucleosynthesis (BBN). The author briefly comment on this but I think it would be important to clarify a few points in that regard:

  1. For the ALP discussion, at what photon temperature is the signal being generated? They hint that there might be mechanisms to avoid Neff constraints towards the end of section 3, but that would likely depend heavily on the photon temperature at which the signal was generated.

  2. In the PT scenario, they conclude towards the end of their analysis that due to the low temperature requirement, the PT should take place in a dark sector. If that is the case, is the temperature in Fig. 3 related to the photon or dark sector's temperature? Does having the transition occur in a dark sector significantly change those figures? I believe that this point should be clarified in the text in a way that one can see at what temperature of the standard model (SM) plasma the effect is taking place.

  3. Towards the end of section IV they say that the energy density in the dark sector should either get transferred to the photon plasma, which requires the transition to occur before 1 MeV, or go to dark radiation which is then constrained by Neff. For the energy transfer to the SM plasma, doesn't it need to occur at temperatures a factor of a few higher than MeV since such transfers are rarely very efficient? For the case in which the energy is transferred to dark radiation, could this allow phase transitions happening at (photon) temperatures below an MeV? If not, wouldn't this imply that the curve with $\beta/H = 100$ in Figure 3 is already ruled out?

Another thing that is missing in their manuscript is a comparison between the two scenarios. Is there already a preference in the data between the ALP and phase transition signals? Assuming the NANOGrav signal is coming from BSM physics, what are the prospect for distinguishing between different scenarios with future data?

In summary, the paper has interesting results and is overall well written, but would benefit from clarifying and expanding on a few important points. After addressing those points I would recommend the manuscript for publication in SciPost Physics.

  • validity: -
  • significance: -
  • originality: -
  • clarity: -
  • formatting: -
  • grammar: -

Author:  Pedro Schwaller  on 2021-01-06  [id 1125]

(in reply to Report 2 on 2020-11-20)

We thank the referee for the positive evaluation of our work. We already addressed the lacking comparison in terms of goodness of fit between the two scenarios in our response to the first referee. Therefore we only comment here on the three points raised in regard to the existing constraints:

  1. Depending on the exact values of the axion mass, coupling strength and misalignment angle the signal is generated at a temperature of several MeV. We agree nevertheless that constructing a mechanism that depletes the dark photon energy fast enough would be quite a model building challenge. Since we don't have a specific mechanism in mind we altered the hint at the end of section 3, such that it reflects these difficulties better.

  2. All temperatures in the text and figures refer to the one of the SM plasma. It is correct that the temperature of the dark sector can differ from the SM plasma. Here we have assumed that at the time of the PT, the temperatures of the visible and dark sectors agree. We have added a footnote to clarify this. The case where the dark sector temperature differs was discussed in detail e.g. in our reference [26]. Reducing the dark sector temperature might reduce the tension with Neff or BBN constraints, however at the same time the GW signal is strongly suppressed with the fourth power of the temperature ratio, which makes fitting the NANOGrav signal challenging.

  3. Relating the bounds from BBN to an exact temperature at which the transition has to occur is of course only possible in a given model. Since we had no firm bounds on the transfer efficiency we decided to show the approximate bound of 1 MeV for the transition temperature. Taking a more conservative bound at a few MeV would however not change our qualitative results.

In the case in which the energy is transferred to dark radiation, the whole dark sector is only interacting gravitationally with the SM. This indeed allows for the possibility of the transition taking place after BBN, however is still subject to constraints from Neff, as discussed above and in Ref. [26]. More precise answers can only be obtained for concrete models, which we plan to investigate in the future.

---

## Round 2 · Referee Report · Anonymous (Referee 1) · 2021-2-1

Strengths

The paper discusses a recent NANOGraph anomaly in terms of several primordial signals. It is clear and well written.

Weaknesses

The discussion is partially agnostic when it comes to concrete models which makes some bounds (like BBN) hard to assess.

Report

The authors amended the issues I had with the previous version. I recommend publication.

---

## Round 2 · Referee Report · Anonymous (Referee 2) · 2021-2-12

Strengths

1- Interesting analysis of potential physics implications of new experimental result with two different models 2- Main ideas and results are clearly presented

Weaknesses

The model agnostic approach to the phase-transition scenario makes it difficult to access the cosmological viability of these scenarios.

Report

The authors have addressed all the points I had brought up in my first report. I recommend publication of the manuscript.

---

## Editorial Decision

published